# Essential Oils as Novel Anthelmintic Drug Candidates

**DOI:** 10.3390/molecules27238327

**Published:** 2022-11-29

**Authors:** Sujogya Kumar Panda, Marijn Daemen, Gunanidhi Sahoo, Walter Luyten

**Affiliations:** 1Department of Biology, KU Leuven, 3000 Leuven, Belgium; 2Center of Environment Climate Change and Public Health, RUSA 2.0, Utkal University, Bhubaneswar 751004, Odisha, India; 3Department of Zoology, Utkal University, Bhubaneswar 751004, Odisha, India

**Keywords:** anthelmintic, essential oil, gastro-intestinal nematodes, human helminths, synergy, toxicity

## Abstract

Helminths, with an estimated 1.5 billion annual global infections, are one of the major health challenges worldwide. The current strategy of the World Health Organization to prevent helminth infection includes increasing hygienic awareness, providing better sanitation and preventative anthelmintic drug therapy in vulnerable populations. Nowadays, anthelmintic drugs are used heavily in livestock, both in case of infection and as a preventative measure. However, this has led to the development of resistance against several of the most common drugs, such as levamisole, ivermectin and thiabendazole. As many as 70% of the livestock in developed countries now has helminths that are drug resistant, and multiple resistance is common. Because of this, novel anthelmintics are urgently needed to help combat large-scale production losses. Prior to this review, no comprehensive review of the anthelmintic effects of essential oils and their components existed. Multiple review articles have been published on the uses of a single plant and its extracts that only briefly touch upon their anthelmintic activity. This review aims to provide a detailed overview of essential oils and their components as anthelmintic treatment against a wider variety of helminths.

## 1. Introduction

### 1.1. Epidemiology of Human Helminthosis

Parasitic worms or helminths are one of the main health challenges in developing countries in Africa, Asia and South America. Each year, approximately 1.5 billion individuals are infected worldwide, with many being host to multiple parasitic species simultaneously [1,2]. It is estimated that up to 420,000 people die annually due to various forms of helminthosis (helminth infection). However, cases of disability are much more common, and the global helminthic burden is estimated at 51.7 million disability-adjusted life years (DALYs) [3,4]. The reason for this high rate of helminthosis is two-fold; the aforementioned regions are characterized by high poverty and poor sanitation, and they generally have tropical and subtropical climates [5,6,7,8]. Children are especially prone to ascariasis and schistosomosis in comparison to adults [9,10]. Helminthosis leads to growth deficits, impaired memory and reduced educational performance among children [4,11].

Knowledge of helminthosis goes back to the Roman Empire and early Arab scholars, but a true understanding of helminths’ biology was only attained starting from the 18th century [12,13]. Today, the most common helminthoses are part of the neglected tropical diseases (NTDs). These are afflictions that are linked to poverty-stricken areas in tropical and subtropical environments but are not at the center of international efforts [14]. Sub-Saharan Africa is particularly heavily afflicted by NTDs, of which helminthoses form the vast majority of cases [15]. Eight out of twenty NTDs, as defined by the WHO, are caused by helminths [16,17]. Preventive deworming, in addition to improving hygiene and sanitation, are the main WHO-recommended interventions [2,14,18]. Thanks to these interventions, a decrease in infections has been observed in the period from 2006 to 2016. However, there are still several challenges in treating helminthosis, including recurrent infections, “super-spreaders”, persistence despite treatment and polyparasitism [19]. Diagnosis is often difficult and usually occurs by detection of eggs or larvae passed in feces [20]. In 2012, the signatories of the London Declaration vowed to eradicate Guinea worm disease, eliminate lymphatic filariasis and control schistosomosis, soil-transmitted helminths and onchocerciasis by 2020 [21]. Despite WHO efforts, investment in treatment and prevention remains low, and calls have been made to strengthen the strategy [22].

There are an estimated 76,930 species of parasitic helminths, with 43,945 known host species [23]. However, cryptic diversity, especially among trematodes, makes it difficult to assess the true number of species [24]. Three hundred species of helminths are capable of infecting humans, although accidental infection by species with different host specificity is possible [12,13,25]. In addition, understanding the factors involved in host shifting is important to limit the spread of newly discovered parasites to humans [26]. A small selection of human helminths is shown in Figure 1. Helminth is a general term, which is applied to multiple species from the phyla Nematoda (roundworms) and Platyhelminthes (flatworms). Parasitic flatworms belong to either the class Cestoda (tapeworms) or Trematoda (flukes). Within the Nematoda, parasites can be found in both classes Chromadorea and Enoplea [12,27,28,29].

Prior to this paper, no comprehensive review of the anthelmintic effects of essential oils and their components existed. Multiple review articles discuss the uses of a single plant and its extracts and only briefly touch upon their anthelmintic activity [30,31,32]. Luna et al. discussed the use of essential oils and their components against neglected diseases but mostly focussed on trypanosomiasis [33]. Another recent review deals with the effects of essential oils and their components on *Schistosoma mansoni* [34]. Under the circumstances, a review describing the anthelmintic effects of essential oils against a wider variety of helminths seems timely.

### 1.2. Common Human Helminths

#### 1.2.1. Nematoda

The most widespread human helminths belong to the phylum Nematoda. They are divided into two groups: the soil-transmitted nematodes (afflictions include: ascariasis, trichuriasis, hookworm infections, enterobiasis and strongyloidiasis) and the filarial nematodes (causing lymphatic filariasis, onchocerciasis, loiasis and dracunculiasis). Several anatomical traits distinguish nematodes from platyhelminths, of which the cylindrical body shape and an exoskeleton (cuticle) are the foremost. Due to this exoskeleton, nematodes must undergo four larval molts before transitioning into adulthood. The parasitic nematodes are typically dioecious, with both male and female forms, although *Strongyloides stercoralis* can reproduce by parthenogenesis [1,27,35].

There are some variations among the life cycles of various soil-transmitted helminths. *Ascaris lumbricoides*, *Trichuris trichiura* (whipworm) and *Enterobius vermicularis* (pinworm) infect through the ingestion of infective eggs. In contrast, hookworms (*Ancylostoma duodenale* and *Necator americanus*) and *Strongyloides stercoralis* infect by means of free-living larval stages that hatch from eggs excreted in fecal matter. *E. vermicularis* is unique in that females migrate out of the anus to lay eggs on the perianal skin [27,36]. Globally, the soil-transmitted helminthoses are responsible for more than 1.5 billion infections, up to 135,000 deaths and upwards of 39 million DALYs [3]. Ascariasis, trichuriasis and hookworm infections are the most common afflictions caused by parasitic nematodes worldwide [1,3,17]. In the developed world, pinworm infection is the main type of soil-transmitted helminthosis [37].

Filarial nematodes are less widespread. The most common infections are lymphatic filariasis (*Wuchereria bancrofti* and *Brugia malayi*), onchocerciasis or river blindness (*Onchocerca volvulus*), loiasis (*Loa Loa*) and dracunculiasis or Guinea worm disease (*Dracunculus medinensis*) [1]. Filarial nematodes require an arthropod intermediate host, such as a mosquito or a blackfly, with the exception of *D. medinensis*, which is not a true filarial nematode [27]. Lymphatic filariasis and onchocerciasis are, respectively, responsible for 120 million and 21 million infections worldwide and for a total burden of 7.3 million DALYs. *Wuchereria bancrofti* is the leading cause of lymphatic filariasis (90% of cases) and has been confirmed to be endemic in 76 countries [38]. Death is uncommon; instead, both diseases cause high morbidity in the form of elephantiasis and blindness, respectively [3,39,40]. An estimated 44 million people suffer from overt morbidities connected to *W. bancrofti* [38]. Guinea worm disease is on the verge of eradication, with only 54 cases reported in 2019 [41].

#### 1.2.2. Platyhelminthes

##### Trematoda

The trematodes that are infectious to humans can be divided into four groups: liver flukes (*Clonorchis sinensis, Fasciola hepatica*, *F. gigantica*, *Opisthorchis viverrini* and *O. felineus*), lung flukes (*Paragonimus* spp.), intestinal flukes (*Fasciolopsis buski*) and blood flukes (*Schistosoma mansoni*, *S. japonicum*, *S. mekongi*, *S. haematobium* and *S. intercalatum*) [12,27,42,43]. Liver, lung and intestinal flukes are labeled as foodborne trematodes by the WHO. They can be acquired by consuming undercooked fish or watercress, crustaceans and water chestnuts, respectively. All three types use snails as intermediate hosts. After hatching, the miracidia (larvae) infect aquatic snails through ingestion or penetration and form cercariae. These cercariae migrate to the environment and encyst either within a second host (fish or crustaceans) or on a plant surface, depending on the species. The cysts (metacercariae) are then ingested by the final human host and develop into hermaphroditic adults in either the liver, lungs or the intestines. Eggs are passed in the sputum (lung flukes), stool or urine [27,44]. Annually, foodborne trematodes infect 200,000 people, cause 7000 deaths and create a burden of more than 2 million DALYs [42]. People who work in aquaculture are especially at risk due to contact with fresh water [45]. The blood flukes of the genus *Schistosoma* are responsible for the disease schistosomosis (bilharzia). They are not transmitted via food; instead, the cercariae directly penetrate their final host, often via the foot. They develop into adults in the lungs and the hepatic portal system before migrating to the venules of the intestine or the bladder. Schistosomes are dioecious, and the smaller female resides in a tegumental fold on the ventral side of the male [27,46,47]. In 2018, 290.8 million people required treatment for schistosomosis [43]. The global burden of the disease has been estimated at 1.7 million DALYs. However, these do not take related morbidities, such as cirrhosis and cancer of the bladder and liver, into account [48].

##### Cestoda

Tapeworms have several unique anatomical features. They do not possess an alimentary canal; instead, nutrients diffuse through the integument. In addition, they possess a scolex or head, which is connected via a neck to a series of repeating segments called proglottids. The proglottids at the neck get detached from the posterior end when mature. These terminal segments carry eggs that are passed in the stool upon release. Upon ingestion by the intermediate host, the eggs hatch into oncospheres, which migrate to the muscles and develop into cysticercus larvae (or hydatid larvae, depending on the species). The cysticercoid larvae mature into adults in the intestine of the definitive host [27].

Approximately 40 species of tapeworms are capable of infecting humans, although most of these infections are accidental. *Taenia saginata* (beef tapeworm) and *T. solium* (pork tapeworm) can live in the intestines as adults in a condition called taeniasis. However, infection by the cysticercus stage is also possible after consumption of eggs, leading to cysticercosis [12,49]. Taeniasis causes few clinical symptoms. In contrast, cysticercosis can cause blindness, epileptic seizures and convulsions due to formation of cysts in the brain (neurocysticercosis). The number of people affected by neurocysticercosis worldwide is estimated between 2.56 and 8.30 million, many of whom receive poor treatment [49].

*Echinococcus* is another genus of tapeworm capable of infecting humans. They can infect humans as an intermediate host. The disease is called echinococcosis or hydatidosis and is characterized by the formation of hydatid cysts in the liver and lungs. More than 1 million people suffer from echinococcosis worldwide [27,50].

### 1.3. Animal Parasites

Helminth infections are not solely limited to humans but are widespread among all vertebrates. Of the estimated 76,930 helminth species, 1555 infect cartilaginous fish, 14,154 infect bony fish, 4225 infect amphibians, 11,486 infect reptilians, 33,849 infect birds, and 11,631 infect mammals [23]. Of special economic and societal importance are those helminth species that infect livestock. Helminthoses are a major cause of livestock production loss worldwide [51], the main reasons being direct damage caused by helminths, energy use for immune defense and reduced food intake by infected cattle. These manifest themselves in the form of reduced milk production, carcass weight and reproduction [52]. The worldwide impact is difficult to assess, but gastro-intestinal nematodes are the cause of the livestock diseases with the greatest impact on the poor [53]. Common veterinary helminths are the nematodes *Ostertagia ostertagi* (cattle), *Haemonchus contortus* (sheep and cattle), *Teladorsagia circumcincta* (sheep and goats), *Dictyocaulus viviparus* (ruminants), *Ascaris suum* (pigs) and the trematode *Fasciola hepatica* (sheep and cattle) [52,54]. Aside from food production, food safety is also a concern, as livestock can act as intermediate hosts for some infections. For example, the tapeworms *T. saginata* and *T. solium* are transmitted to humans through the ingestion of undercooked meat [12]. Some helminths, such as the liver flukes *F. hepatica* and *F. gigantica*, are able to infect livestock as well as humans [53]. *Ascaris suum*, the large roundworm of pig, might be the same species as *A. lumbricoides*, the foremost human roundworm parasite [55]. Treatment of these helminths in livestock could help limit future infection in humans. Climate change will likely cause a shift in the prevalence and intensity of helminthoses. In order to maintain food production, adapted preventative and treatment measures are necessary [56,57].

Detection of these parasites is often difficult, as many infections are asymptomatic. In the past, preventive treatment with anthelmintics was used, but this led to increasing anthelmintic resistance [52]. Infective *Ostertagia* larvae can remain on pastures for up to a year, indicating that long-term treatments may be needed to fully eliminate helminths from a herd [58]. Many helminths have a broader host specificity; hence, wild ruminants can often infect domesticated cattle. Wildlife could thus be a possible source of reinfection even if the diseases have been eliminated previously [59]. 

In addition to anthelmintic treatment, there are several strategies to limit infection among livestock. Simple practices, such as grazing management, biological control and supplementary feeding, can be followed without the need for additional research. Grazing management can help limit the exposure of livestock to helminths. Letting livestock sequentially graze on different plots causes the number of infective helminths to decrease due to the absence of their host before the cattle are reintroduced. Biological control involves denying helminths the necessary environment for infection. The most logical procedure is through dung removal, but supporting wildlife can also lead to decreased infection. For example, ducks can feed on snails and deprive *Fasciola* spp. of their intermediate hosts [60]. Additionally, soil saprophytic fungi, viz. *Mucor circinelloides*, *Pochonia chlamydosporia*, *Duddingtonia flagrans*, *Arthrobotrys* spp. and *Monacrosporium* spp., can act as effective biological anthelminthic agents to reduce the presence and survival of free-living stages (eggs, larvae), thus avoiding infection. The health of the infested animals can also be improved through supplementary feeding. The genetic resistance of livestock can also be increased through selective breeding and crossing. Maasai sheep have increased resistance against nematodes, and trematode resistance has been observed in Javanese thin-tailed sheep. Crossing these breeds with high performance breeds can lead to decreased production loss [61]. Finally, the possibility of helminth vaccines has been discussed for several decades. As of yet, no large-scale veterinary (or human) vaccine programs exist. However, research into vaccines for *H. contortus* has been underway for many years, and recent human vaccine trials against hookworms increase the prospect of veterinary vaccines [60,61,62,63]. The eventual end goal would be a multivalent vaccine that protects against several helminths, including *Haemonchus* spp. and *Ostertagia* spp. [53]. 

## 2. Current Anthelmintics

Anthelmintics are drugs used for the treatment of parasitic worms in humans, animals or plants. While the vast majority act upon receptors of the neuromuscular system in helminths, the efficacy differs strongly between nematodes, trematodes and cestodes [64]. The anthelmintic strength often differs even among the more closely related gastro-intestinal nematodes [65]. An overview of some of the major anthelmintic drugs follows.

### 2.1. Ivermectin

Ivermectin, a macrocyclic lactone and derivative of avermectin, is one of the most potent anthelmintics used in both human and veterinary therapies. In humans, it is used as a treatment for filariasis and onchocerciasis, whereas it sees broader antinematodal use in livestock and domestic pets [66,67,68]. Ivermectin allosterically activates glutamate-gated chloride channels (GluCl), resulting in hyperpolarization of the cell and paralysis in the pharyngeal and body wall muscles of the nematode [64,69]. Resistance to ivermectin has been observed in *Cooperia* spp. and *Ostertagia* spp. [66,70,71]. In the Netherlands, as many as 78.3% of sheep flocks harbor ivermectin-resistant parasites [72]. The genetic basis for resistance has been identified in *Caenorhabditis elegans*. Mutations in the GluCl subunits *glc-1*, *avr-14* and *avr-15* can decrease susceptibility. Additionally, increased expression of P-glycoproteins leads to increased resistance against ivermectin, as well as moxidectin, levamisole and pyrantel [64,69,73,74]. However, it is unclear whether resistance in parasitic nematodes has the same mutational basis [66]. Aside from the issue of resistance, avermectin excreted in feces may be harmful to larvae of coleopterans [75].

### 2.2. Nicotinic Acetylcholine Receptor (ant)Agonists

Nicotinic acetylcholine receptors (nAChRs) are targeted by a variety of anthelmintic drugs with differing chemical structures. Levamisole (an imidazothiazole), pyrantel and morantel (both tetrahydropyrimidines) act as agonists on L-nAChRs (levamisole-sensitive nAchRs), mimicking the action of acetylcholine. They cause prolonged excitation, leading to muscle spasms and paralysis [69]. They are used to treat intestinal nematodes in humans, although levamisole has a low therapeutic efficacy [64,68]. Resistance to all three drugs has been observed in human and veterinary helminths. Due to their shared mechanisms, single mutations can lead to multiple resistance [68,76]. Levamisole-resistant *C. elegans* mutants have led to the identification of more than 20 mutant genes that confer resistance, several of which have also been observed in parasites [64,69,77].

Derquantel acts as a competitive antagonist of nAChRs and has a preference for bephenium-sensitive B-nAChRs [64]. It has synergistic effects with abamectin, an ivermectin analog, but efficacy of this treatment against *Haemonchus* spp. has begun to decline [69,78]. No modes of resistance are currently known.

Amino-acetonitrile derivatives (AAD) are a relatively new class of anthelmintic drugs [79]. Monepantel, an AAD, targets nAChRs with ACR-23 subunits belonging to the DEG-3 family [64,69]. Resistance has been detected in *C. elegans* through mutations in the *acr-23* gene [80]. There are indications of resistance in *H. contortus*, only four years after its introduction [72,78,81]. Whether the resistance mechanism is similar to that in *C. elegans* is unclear; recent in vivo selection for resistance could provide insight into the exact mechanisms [82]. 

### 2.3. Benzimidazoles

Benzimidazoles include thiabendazole and albendazole among others and are used to treat a wide range of flukes, tapeworms and roundworms. They act by interfering with the cytoskeletal structure through the binding of β-tubulin monomers. This results in decreased intracellular transport with far-ranging effects, ending in paralysis and death [64,69,83]. Drug resistance has been observed in sheep and goat parasites across countries [84,85,86,87]. The resistance mechanism is connected to several possible mutations in the β-tubulin gene at codons 167,200 (both phenylalanine to tyrosine) and 198 (glutamic acid to alanine) [69,87]. Alanine in position 198 produces the strongest resistance but only in combination with the benzimidazole-susceptible single-nucleotide polymorphisms in codon 200 [88]. The phenylalanine-to-tyrosine substitution in codon 200 corresponds to the allele in mammals, likely explaining the specificity [87].

### 2.4. Cyclooctadepsipeptides

Emodepside was developed as a novel anthelmintic effective against levamisole-, ivermectin- and benzimidazole-resistant helminths [64]. It is a broad-spectrum anthelmintic, which affects both gastro-intestinal nematodes and larval stages of filariae. In *C. elegans*, it is thought to act intracellularly on the calcium-activated phosphate channel SLO-1 located in the body wall muscle. The activation results in increased opening of the channels, and as a result, a decrease in locomotion and feeding behavior [89]. It has been proven to be effective in a wide range of hosts, including sheep, cattle, rodents and reptiles [90,91].

## 3. Future Strategies

The discovery of novel anthelmintic drugs is advancing slowly, despite the severity of human helminthoses. The main drive behind anthelmintic research are veterinary applications, and many drugs used in humans today were originally developed for livestock [64].

Anthelmintic resistance is a widespread and growing phenomenon, especially in small ruminants and, to a lesser but increasing degree, in cattle [76,92]. Resistance occurs worldwide but is more pronounced in the southern hemisphere due to longer seasonal activity of parasites [53]. As of now, the efficacy of ivermectin, levamisole and benzimidazoles has decreased considerably [64,93]. Helminths’ resistance lowers the productivity of livestock and endangers the sustainability of the current production levels [92,94]. Factors contributing to the development of anthelmintic resistance include large-scale prophylactic use, high frequency of treatments, single-drug treatments and underdosing [52,68,95]. Several improvements to current practices could serve to limit the rise and spread of resistant strains. In addition to limiting infection, grazing management and biological control could serve to limit the spread of resistant strains [60,76]. Targeted intervention, instead of drenching, and fewer treatments, are preferred to limit unnecessary exposure of helminths to anthelmintics. To limit survivability, correct dosages should also be used [68,96]. A combination therapy is also recommended. Although multiple resistance is not uncommon, using multiple drugs simultaneously would likely result in better treatment results. This is partly due to the absence of multiple resistance but also due to possible synergistic effects [68,69,76,97].

Whether anthelmintic resistance is present in human helminths is unclear, but the possibility should be considered seriously, and efforts should be undertaken to limit it [68,98,99].

Aside from possible methods to limit anthelmintic resistance, there is an urgent need for novel anthelmintics [93]. Several helminthoses are poorly treatable with currently available drugs, requiring new targets to be explored [68,100]. By using genome-wide approaches, it could be possible to identify putative anthelmintic targets and ways in which resistance can occur [101]. Targeting drugs against multiple members of a multigene family can also limit the development of resistance [73]. Screening compound libraries as well as examining the repurposing of existing drugs could lead to new candidate compounds [98,102]. A possibly interesting avenue for novel drugs is neuropeptides. These have an important role in nematode signaling but have not been the target for anthelmintics as of yet [99].

## 4. Anthelmintic Assays

The most straightforward method to test anthelmintic activity is the fecal egg count reduction test—an in vivo assay that compares the number of helminth eggs in the feces between treated and untreated groups [103]. Large-scale in vivo tests like this are impractical and unethical due to the number of animal hosts necessary [93]. The adult stage is the most harmful and clinically meaningful in helminth infections. Unfortunately, there are no ways to test in vitro anthelmintic activity against adult parasitic helminths that do not require post-mortem collection from hosts. Anthelmintic assays, therefore, require either larval stages of helminths or a model organism, such as *C. elegans*, to measure anthelmintic activity [104]. Such assays typically measure easily quantifiable traits, such as development, growth, behavior, motility or lethality [64]. Various assay designs are possible based on the helminth life cycle—egg hatch assays, larval motility assays and adult motility (and mortality) assays being the most common [104,105]. Egg hatch tests can be conducted directly on parasitic helminths via eggs collected from feces [106,107], as well as in *C. elegans* [105]. They typically consist of egg collection, compound addition and counting of the ratio of unhatched eggs [103,106]. However, this is impractical and labor intensive [93]. Similarly, larval motility assays require collection of infective larvae from feces. Anthelmintic activity is then estimated by observing the larval movement across a sieve or surgical gauze, which is placed on or in the liquid medium [108,109].

Adult (and L4) motility and mortality assays can take several forms [98]. The simplest readout is determining the motile/immobile or dead/alive ratio among the worms. Though quite robust, this method is labor intensive [104]. There are also automated alternative measurements. WormScan uses light stimuli to induce negative phototaxis in *C. elegans* in order to assess motility. Two sequential scans are performed, which allows the identification of immobile or dead worms [110,111]. Recently, an automated version, aptly called Automated WormScan, was developed, which generates images based on the detected differences between the scans. This allows for high throughput with minimal operator time required [112]. A third method is the WMicrotracker (Phylumtech, Argentina). It utilizes infrared beams and detects beam interruptions caused by the movement of *C. elegans* in a microtiter plate. These fluctuations in the signal intensity are used as a measure of the locomotor activity [113,114].

Positive hits from the aforementioned assays can be further tested in egg-hatching assays of parasitic nematodes as an additional test of relevance [115]. The assays can also be applied to anthelmintic mechanism discovery via forward genetics approaches. Random mutagenesis screens with ethyl methane-sulfonate mutants can be subjected to the assay to establish the degree of anthelmintic resistance and to identify putative targets [64,110,115]. Testing mutants resistant to the current anthelmintics can also help select compounds with novel targets [115]. Additional information can be obtained through morphological analysis with scanning electron microscopy and transmission electron microscopy [105,116]. 

## 5. Essential Oils as Anthelmintic Candidates

Essential oils (EOs) are mixtures of volatile hydrophobic secondary plant metabolites of low molecular weight. They are usually extracted from plants through steam distillation and more rarely by cold pressing. Chemically, they are blends of up to hundreds of different plant metabolites [117] and include aromatic alcohols, acids, esters, phenolics, ketones, aldehydes and hydrocarbons. EOs usually contain two to three major terpene or terpenoid components, which constitute up to 30% of the oil [117]. EO-derived bioactive molecules, such as terpenoids and phenylpropanoids [118,119], are widely used in pharmaceutical sciences, medicine, biology and agronomy. EO compounds may have antitumor, larvicidal [120], insecticidal [27,121] or anthelmintic activities [105], as well as activity against arbovirus vectors [122,123]. The Arabs used EOs since the Middle Ages against a variety of pathogens. Bakkali et al. reviewed the biological effects of Eos, including their antimicrobial, anti-inflammatory, analgesic, spasmolytic, sedative and local anesthetics properties [117]. 

Due to their promising biological effects, EOs and/or their derived components have gained much attention nowadays [124,125,126]. The lipophilic nature of essential oils allows them to cross the membranes of parasites, as well as the blood–brain barrier, opening possibilities to combat the second stage of several of these infections [33]. Generally, EOs can induce oxidative stress in parasites, increasing the levels of nitric oxide in the infected host, reduce parasite resistance to reactive oxygen species and increase lipid peroxidation, ultimately leading to serious damages to cell membranes [33]. Reviews on EOs are available for the control of veterinary ectoparasites [127], neglected tropical diseases and arboviruses [33], and infections with protozoa or helminths [128]. We conducted an exhaustive search for literature describing the anthelmintic effects, either in vitro or in vivo, of EOs and their principal components (see Table 1 and Table 2). The search terms “(essential oil AND helminth) OR (essential oil AND anthelmintic)” were used to query PubMed (for details, see Appendix A). We found 63 articles that described anthelminthic assays of EOs or EOCs. Figure 2 describes the distribution of these papers by year. An increased interest starting from 2011 is visible in our results, with a peak in 2019. The range of helminth or model species used in anthelminthic EO research is shown in Figure 3, with some studies utilizing more than one species. The majority of articles studied species belonging to the phylum Nematoda (45 out of 63 articles), with *H. contortus* (24 articles) as the most utilized helminth species. Other Nematoda are unspecified gastro-intestinal nematodes in in vivo research (nine articles) and *Anisakis simplex* in research on fish parasites (four articles). The anthelminthic effects on Trematoda are studied in 13 articles: 12 on *Schistosoma mansoni* and 1 on *Schistosoma haematobium*. *Echinococcus* spp. belonging to the phylum Cestoda are studied in 10 articles. Monogenean *Gyrodactylus* spp. and the Annelid *Pheritima posthuma* are studied in one article each. The range of assays used in both in vitro and in vivo studies is shown in Figure 4. In in vitro studies, the most used methods are egg hatch assays (23 articles), mortality assays on adults (16 articles), larval development assays (15 articles) and protoscolex viability assays conducted on *Echinococcus* spp. (10 articles). In vivo studies depend heavily on fecal egg count reduction assays (10 articles) and measurements of the animal’s worm burden (8 articles).

## 6. Chemical Classes of EO Components with Reported Anthelmintic Activity

### 6.1. Monoterpenes

Limonene is a major terpene (28% to 73%) in EOs from *Eucalyptus staigeriana*, *Citrus reticulata, C. limonia* and *C. aurantifolia* [137,138,145,152]. It is effective at variable EC_50_ against *H. contortus*, a pathogenic nematode of ruminants, as determined by egg hatch assays and larval development assays [138]. These authors have also ascertained its effects in goat and sheep models (500 mg/kg) through the fecal egg count reduction assay [137]. Its efficacy has been proved for *Schistosoma mansoni*, causing full inhibition of motion at 43.75 µg/mL and with an LC_50_ of 81.7 µg/mL [152,160].

Another monoterpene, gamma-terpenene, has been isolated as a major component (10 to 35%) of *Zanthoxylum zanthoxyloides*, *Melaleuca alternifolia* and *Bunium persicum* EOs [140,151,162]. It is present along with undecane (14.84%), carvacrol (35%), terpinen-4-ol (41.98%) and cumin aldehyde (15.5%). It is effective against hydatid cysts of *Echinococcus* spp. (3.125 to 50 µL/mL) [151], assessed through the protoscolex viability assay, and against *H. contortus*, determined via the egg hatch assay (LC_50_ = 430 µg/mL) [179].

*p*-Cymene (synonym: *p*-cymol or *p*-isopropyltoluene) is an alkyl-substituted aromatic compound. Although not a major constituent, it is present in EOs of many plants, such as *Origanum compactum*, *Alpinia zerumbet*, *Thymus vulgaris*, *Trachyspermum ammi* and *Nigella sativa* [141,143,163,166]. It is effective against *Echinococcus* spp. and *H. contortus*. This compound is well known for its health benefits and antiparasitic properties [180].

α-Pinene (24 to 27% in *Myrtus communis* and *Pinus nigra*) is active against hydatid cysts of *Echinococcus* spp. assessed through protoscolex viability assays, where 6.3 ± 0.3% and 100% reduction occurred after, respectively, with 5 to 30 min of exposure [161,168]. β-Pinene is a major constituent in EOs of plants, such as *Artemisia campestris* (36.4%) and *Ferula gummosa* (57.0%), and is effective against *Heligmosomoides polygyrus* [129] and *Echinococcus* spp. [158]. It is also a minor constituent in *Citrus aurantifolia* EO, along with others (limonene (56.37%), β-pinene (11.86%) and γ-terpinene (11.42%)), and was effective against *H. contortus* [145].

Monoterpene myrcene (β-myrcene) is a minor constituent in the EOs of *Mentha piperita* and *Croton zehntneri*. The compound is well known for its biological properties, as well as its safety (used in the food industry). In several animal studies, β-myrcene has shown promising health benefits. Along with other constituents, it was active against *Trichostrongylus* spp. and *H. contortus* when tested in a larval development assay [106].

### 6.2. Sesquiterpenes

Sesquiterpenes are terpenoids (C15) containing three isoprene units and are the second most important group of compounds in plant EOs. Several sesquiterpenes have been reported as antiprotozoal since the discovery of artemisinin [181]. β-Caryophyllene is present in the EO of plants, such as *Bidens sulphurea* (10.23%) [153] and *Newbouldia laevis* (36.08%) [170], and is effective in a mortality assay on *Schistosoma mansoni*. In addition, it is found in the EO of *Plectranthus neochilus*, which is effective against *Strongyloides ratti* [162], as well as in *Pinus nigra* Arn. subsp. *pallasiana* (active against *Echinococcus* spp.) assessed through a larval development assay [168]. Trans-Caryophyllene, found in several EOs, such as *Baccharis trimera*, *Mentha x villosa* etc. [150,160], is effective against *Schistosoma mansoni.* (E)-Caryophyllene, present in variable amounts in several EOs, such as those from *Ageratum conyzoides* and *Lippias idoides,* is effective against several helminths, such as *Trichostrongylus* spp., *Haemonchus* spp., *Syphacia obvelata* and *Aspiculuris tetraptera* [130,139]. Caryophyllene oxide (50.26%) and copaene (10.58%) in *Lantana camara* are also effective in an egg-hatching assay with *H. contortus* [143].

### 6.3. Alcohols

Citronellol or dihydrogeraniol, present in the EOs of *Eucalyptus citriodora* and *Pelargonium roseum* (37.7%), is a natural acyclic monoterpenoid. It is effective against hydatid cysts of *Echinococcus* spp. assessed through protoscolices viability assays measured after 60 min of exposure [158].

Geraniol is the principal component in many EOs of the genus *Cymbopogon.* Its concentration varied from 53 to 81% in the EO of lemon grasses, such as *Cymbopogon citratus*, *C*. *schoenathus* and *C. martinii*. The in vivo efficacy has been tested by Katiki et al. [133] with the parasite *H. contortus* in sheep through fecal egg count reduction, egg hatch and larval development assays. The in vitro effects were tested through fecal egg count reduction, egg hatch and larval development assays against *Trichostrongylus* spp. (CL_50_s = 40–160 µg/mL) [155] and *Pheritima posthuma* (20 mg/mL) [156]. Geranial (18.98%), along with other EO constituents, such as 3,7-nonadien-2-one, 4,8-dimethyl (24.86%) and neral (17.77%), at a dose of 500 mg/kg, was able to reduce the worm burden in sheep [131]. Menthol (42.5%) and menthone (27.4%) from *Mentha piperita* EO were found to be effective against *Trichostrongylus* spp. (egg hatch assay, CL_50_ = 260 µg/mL) [155]. Parreira et al. [149] observed 100% mortality of *Schistosoma mansoni* at 10 µg/mL EC_50_ using *Baccharis dracunculifolia* EO, which contains (E)-nerolidol (33.51%) and spathulenol (16.24%).

Terpinen-4-ol is the main bioactive component of tea-tree oil. It is active against *Anisakis simplex* (up to 10 µL/mL, LD_50_, mortality assay) [182] and *H.contortus* (LC_50_ = 630 µg/mL, egg hatch assay) [179].

β-Linalool, the major component (73.21%) of *Coriandrum sativum* EO, is active against *H. contortus* at 630–2890 µg/mL based on egg hatch and larval development assays [143].

### 6.4. EO Phenols

Thymol, also called 2-isopropyl-5-methylphenol (IPMP), is a natural colorless crystalline mono-terpenoid phenol derivative of cymene, C_10_H_14_O. It is abundantly (50–60%) found in the oils of plants, such as *Melaleuca alternifolia*, *Trachyspermum ammi, Thymus vulgaris* and *Lippia sidoides,* has a pleasant aromatic odor and strong antiseptic properties. It was the treatment of choice for hookworm infection in the United States after 1910 [183,184]. Based on its EC_50_s (390 to 2970 µg/mL, egg hatch assay), IC_50_s (22–497µg/mL, egg hatch, larval motility and larval development assay) and LD_50_s (291 µg/mL, mortality assay), it is highly effective against *H. contortus* [130], *Anisakis simplex* [165], *Syphacia obvelata* and *Aspiculuris tetraptera* [130], *Haemonchus* spp., and *Trichostrongylus* spp., gastro-intestinal nematodes [139] and hydatid cysts of *Echinococcus* spp. [177]. Worm burden, fecal egg count reduction and protoscolex viability assay were also used by various authors.

Carvacrol, isomeric to thymol, is a major monoterpene phenol in the EO of several plants, especially *Zanthoxylum simulans, Satureja montana* (40–64%), *S. khuzistanica* (94%), *Origanum dictamnus* (44–50%), *O. compactum* (50–59%), *Thymus caespititius* (54%), *Thymbra capitata* (68%), *Origanum vulgare* (68.5%), *Mentha spicata* (64.5%) and *O. syriacum* (82%), etc. This compound is active against *H. contortus* (EC_50_ = 320 µg/mL, egg hatch assay) [146], *Anisakis simplex* (LC_50_ = 87 µg/mL, larval mortality assay) [167], hydatid cyst of *Echinococcus* spp. and protoscolices (3000, 5000 and 10,000 µg/mL through a protoscolex viability assay after 10 and 60 min exposure) [173] and *Schistosoma haematobium* (1 µg/mL, mortality test; the minimum concentration for immobilization was determined using an ethyl-acetate extract with this compound as a major constituent) [166].

### 6.5. Phenyl Methyl Ethers

Eugenol, an allyl chain-substituted guaiacol, is a significant component (43.7 to 53%) of the EO of *Ocimum basilicum* and other plants, such as clove. Its efficacy was assessed through mortality and egg hatch assays, and it has been proven with *H. contortus* (662.5 to 5.300 µg/mL). According to El-Kady et al. [185], there are many activities and properties of eugenol that remain undiscovered and need to be further investigated to elucidate the anthelminthic properties of eugenol, both in vivo and in vitro [185]. Anethole (1-methoxy-4(1-propenyl)-benzene) constitutes up to 70% of the EOs of *Foeniculum vulgare* and is effective (10 to 100 µg/mL) against *Schistosoma mansoni* (a water-borne blood fluke of humans) based on mortality tests [159]. It is also a minor constituent in EOs of *Croton zehntneri* and effective against *H. contortus* [130]. Estragole (an isomer of anethole) is a major component (72%) of the EO of *Ocimum basilicum.* It is mostly active against GI nematodes (cattle, 11,100–21,590 µg/mL, LC50, larval mortality and larval migration assays) [106].

### 6.6. Aldehydes

Citral (a natural mixture of geranial and neral) is the second most effective drug after carvacrol against *Haemonchus contorts* parasites at concentrations of 130 to 1370 µg/mL (EC_50_, egg hatch and larval development assays) [132].

Citronellal is a monoterpenoid aldehyde, which is the main component (around 67%) in the mixture of terpenoids that give citronella EO its distinctive lemon scent. It’s in vitro efficacy against *H. contortus* is 410 to 1700 µg/mL (EC_50_, motility assay) [134]. When tested with in vivo models, such as sheep, through a fecal egg count reduction assay for GI nematodes [135], it is effective at 55.9% and 34.5%.

### 6.7. Esters

Esters are also common in EOs due to their widespread occurrence in nature. Linalool acetate (35.97%), along with trans-sabinene hydrate (29.17%), in *Lavandula officinalis* EO was effective in an egg-hatching assay against *H. contortus* (IC_50_ = 316 µg/mL) [145]. Geranyl acetate in *Tetradenia riparia* EO is also effective against *Anisakis simplex* [186]. Isomenthyl acetate (10.1%), a minor constituent along with geraniol (81.4%) in *Cymbopogon martini* EO, is effective against *Trichostrongylus* spp. when tested in egg hatch and larval development assays [155]. Isobutyl angelate (29.26%) and isoamyl angelate (15.27%) from *Anthemis nobilis* EO are effective against *H. contortus* in egg hatch and larval development assays [145]. Citronellyl formate (11.0%), along with another major constituent, such as citronellol (37.7%) and geraniol (17.6%) in *Pelargonium roseum* EO, is effective against *Echinococcus* spp. [158].

### 6.8. Ketones

Monoterpenoid dihydrocarvone is a key building block for synthesizing sesquiterpenes. In *Anethum graveolens* EO, dihydrocarvone (39.1%), along with carvone (22.24%) and D-limonene (16.84%), was effective against *H. contortus* at IC_50_ = 6 µg/mL, ascertained through egg hatch and larval development assays [144].

Β-Thujone (84.13%), a monoterpene ketone found in *Tanacetum vulgare* EO, is effective against *Schistosoma mansoni* in a mortality assay (100–200 µg/mL) [175]. This compound is well known for its presence in other genera of *Salvia* and *Artemisia* and is known to be toxic.

### 6.9. EO Quinones

Thymoquinone is also a monoterpene (2-methyl-5-isopropyl-1, 4-benzoquinone) found in the EOs of plants, such as *Lippia sidoides* and *Nigella sativa* (42%), and is active against hydatid cysts of *Echinococcus* spp., as has been demonstrated by Mahmoudvand et al. [163] through protoscolex viability assays (100–10,000 µg/mL) after 10 and 60 min of exposure.

### 6.10. Miscellaneous EO Compounds

1,8-Cineole (eucalyptol) is a major component of EOs (17 to 42%) of several plants, including *Eucalyptus*. This compound is active against *H. contortus* and other GI nematodes (sheep). Camphor (16.65%) and 1,8-cineole (34.56%) are active against *H. contortus* [147]. 1,8-Cineole (24.69%), along with p-cymene (22.56%) and 4-terpineol (17.43%) in *Alpinia zerumbet* EO [143] and 1,8-cineole (42.11%), 2-bornanone (16.37%) and α-pinene (14.76%) in *Rosmarinus officinalis* EO [171], are effective against GI nematodes (sheep). α-pinene (24.7%), 1,8-cineole(19.6%) and linalool (12.6%) from *Myrtus communis* EO are effective against *Echinococcus* spp. [161]. 1,8-Cineole (32.71%), along with eugenol (43.7%) derived from *Ocimum gratissimum* EO, is effective against *H. contortus* in an egg hatch assay [164]. Caryophyllene oxide (50.26%) and copaene (10.58%) in *Lantana camara* EO are also effective in egg-hatching and larval development assays against *H. contortus* [143].

Borneol (18.61%) was more effective than β-elemene (10.87%) in *Zanthoxylum simulans* EO in an egg-hatching assay with *H. contortus* [178]. Piperitenone, derived from the EO of *Tagetes patula* (23.5%), is active against *H. contortus* (40 µg/mL, EC_50_, egg hatch assay) [174]. Gaínza et al. [169] found dillapiole (76.2% in the EOs of *Piper aduncum*) effective against *H. contortus* (100 to 5720 µg/mL, IC_50_). De Melo et al. [142], using a mortality assay, verified the activity of precocene I, a major component (74.30%) of *Ageratum conyzoides* EO, present along with (E)-caryophyllene (14.23%). Rotundifolone (70.96%) is active against *Schistosoma mansoni* at 10 to 100 µg/mL in a mortality assay [160].

It appears from the above that the efficacy of EOs is quite variable. There are differences among the strains, the concentrations used and the assay methods. Significant differences have been observed in EC_50_ or IC_50_ values even when tested on the same species based on the chemical class of the components (monoterpene or phenylpropanoid) or based on the main functional groups (aldehyde, ketone and alcohol). It seems that the carbon backbone is important, with cyclohexene-containing compounds differing significantly from both the phenylic compounds and the alkenes. The difference between phenylic compounds and those with an alkene structure was also observed among the tests. This suggests that the overall structure of the compound is more important for bioactivity than the specific functional groups or other properties. It is also interesting to note that the effective dose may differ between the isomers, suggesting an interaction with a specific drug target (rather than a physicochemical effect on membranes, as is often assumed).

## 7. Traditional Knowledge: The Key to Novel Anthelmintic Drug Candidates

EOs are complex mixtures of substances, which are volatile in nature; they have been used in traditional medicine for millennia. These EOs and their EOC are known to possess many bioactivities, but in the past two decades, researchers have devoted special attention toward their anthelmintic properties. Because of their lower toxicity and frequent use by local practitioners for many parasitic infections in animals and humans, they offer attractive leads for inexpensive and safe drugs.

In Turkish folk medicine, *Pinus nigra* is frequently used for the treatment of worm infection in cattle [187]. This information encouraged Kozan et al. [168] to study its scolicidal activity. The authors conclude that the EO could be a new treatment, as it showed strong inhibition compared to the control group [168]. *Origanum syriacum* is used as vermifuge [188]. Due to its usage in folk medicine, Lopez and coworkers studied the anthelmintic property of its EO and observed strong inhibition against *Anisakis* L3 larvae [165,167]. Accordingly, the EOs extracted from the leaves of *Origanum syriacum* are proposed as an ecofriendly nematocide and support the folk usages of this plant as an antiparasitic [165]. Other authors [141,145] also reported the traditional uses of three plants, e.g., *Citrus aurantifolia, Anthemis nobile* and *Lavandula officinalis*, in folk medicine as anthelmintics. They studied their EOs on different developmental stages of *H. contortus* and concluded that their study validates the ethnopharmacological importance of three EOs as anthelmintics.

*Dysphania ambrosioides* (Chenopodiaceae) is widely used as a vermifuge, and its extract, as well as infusion, were used as an anthelmintic [189]. Its EO was also reported to treat parasitic infections of non-ruminant livestock (e.g., cats, dogs), as well as humans [190]. Because of its frequent use in folk medicine, Soares et al. [157] further investigated the effects of its EO against *Schistosoma mansoni*, finding it 31.8 times more toxic to adult *S. mansoni* worms compared with GM07492-A cells [157]. *Thymus vulgaris* (a plant from the mint family) is used in traditional medicine as an anthelmintic agent [191], which led to further studies of its EO in a *H. contortus* in vivo model [141]. Thymol was found to be the major constituent responsible for the anthelmintic properties, and the authors concluded that this may lead to the development of an anthelmintic drug [141].

The EO of *Foeniculum vulgare* (Apiaceae) is reported in folk medicine as vermifugal [192]. This use in folk medicine led Wakabayashi et al. [159] to study its anthelmintic effects against *S. mansoni*; they concluded that the EO of *F. vulgare* in combination with praziquantel could be considered as an alternative schistosomosis treatment, which can slow down the development of resistance toward praziquantel [159]. *Xanthoxylum bungeanum* (Rutaceae) is another frequently used plant in traditional Chinese medicine to kill intestinal parasites [193]. Therefore, Qi et al. [178] studied its effects on *H. contortus* and concluded that the available information supported the use of this EO as an alternative means to control gastro-intestinal nematodes [178].

*Cymbopogon citratus* (Poaceae) is another shrub reported to have anthelmintic activity in ethnoveterinary practice [194]. Infusions or decoctions of dry leaves are frequently used during schistosomosis [195]. As the EO displayed auspicious results, its use in folk medicine was validated [132]. *Cyperus articulatus* (Poaceae) is used in traditional medicine by tribes in northwestern Cameroon in various ways, e.g., chopped roots and rhizome, decoction with boiling water to treat onchocerciasis. This led to further study on the activity of its roots and rhizomes against the microfilariae and adult worms of *O. ochengi* by Metuge et al. [196], who concluded it to be a potential treatment for onchocerciasis in humans [196]. *Tanacetum vulgare* (Asteraceae) is commonly used in folk medicine as a vermifuge [197], leading to further study on the effects of its EO against *S. mansoni*. The bioactive compounds, as well as its mechanism of action, were elucidated, reinforcing the traditional use of *T. vulgare* as a vermifuge and an anthelmintic [175]. Another plant from Asteraceae is *Cosmos sulphureus* (synonym: *Bidens sulphurea*), which is used to treat malaria in Brazil [198]. The EO of *B. sulphurea* has been evaluated against *S. mansoni* and considered a promising source of a new schistosomocidal drug [153]. Another antiparasitic plant often used in traditional medicine is *Mentha* × *villosa*; its EO and EOC (rotundifolone, limonene, trans-caryophyllene and β-pinene) were tested against adult *S. mansoni*. Rotundifolone was considered a potential source for the development of a new drug against *S. mansoni* [160]. Based on folk medicine use, Moazeni et al. [173] studied the scolicidal activity of the carvacrol-rich EOs of *Satureja khuzistanica*. The activity they demonstrated supports its use in traditional medicine. Another interesting finding supporting the traditional use of two plants, *Zanthoxylum zanthoxyloides* and *Newbouldia laevis*, saw them being selected based on an ethnopharmacology survey, which documented their use by small-scale farmers for treating digestive helminths in small ruminants [162]. The EOs of both plants were active against *Strongyloides ratti***,** both in egg-hatching and larval migration inhibition assays, at concentrations comparable to established anthelmintics (thiabendazole, levamisole) [162]. In Brazilian folk medicine, *Croton zehntneri* and *Lippia sidoides* are used for the treatment of gastro-intestinal diseases [199]. Camurça-Vasconcelos et al. [130] studied the effects of EOs from both plants on intestinal nematodes of mice (*Syphacia obvelata* and *Aspiculuris tetraptera*), as well as in vitro assays on *H. contortus.* Both EOs proved effective, and the authors concluded that further (in vivo) studies are necessary on target species, such as sheep, after determining the absorption and metabolism of these EOs [130].

## 8. Clinical Studies with Essential Oils as Anthelmintic Candidates

Complete or ongoing clinical trials of EOs as anthelmintic agents are lacking. A study by Massoud et al. [200], although too limited to be considered a clinical trial, describes a formulation consisting of 8 parts of *Commiphora molmol* resin and 3.5 parts of its volatile oils being tested on seven individuals exhibiting signs of fasciolosis. The administered dosage regimen (12 mg/kg per day) reduced common symptoms, such as abdominal pain, fever and weight loss, and it reduced fecal egg counts to zero three weeks after treatment [200]. Although mice, goats and sheep are popular models for such studies, most anthelmintic EO research on humans is limited to pre-clinical tests. Several Brazilian studies with EOs from *Eucalyptus citriodora, Eucalyptus staigeriana, Lippia sidoides* and *Thymus vulgaris* [134,135,136,137,138,139,141] provide evidence for in vivo efficacy against sheep and goat gastro-intestinal nematodes similar to ivermectin.

We found two clinical studies with EO components (purchased commercially). Artemether (6 mg/kg every 3 weeks in 5 cycles) was used for schistosomosis [201]. Artesunate-amodiaquinea (4 mg/kg artesunate and 10 mg/kg/amodiaquine) cured *S. mansoni*-infected children and increased their hemoglobin level [202].

## 9. Patent Literature with Essential Oils as Anthelmintic Candidates

We searched for patents using the term “essential oil” in their title or abstract, and the terms “anthelminthic”, “anthelmintic”, “helminth”, “nematode”, “nematocidal” or “antinematodal” in any field through the European Patent Office’s (EPO) online platform Espacenet (https://worldwide.Espacenet.com/patent/ (accessed on 31 May 2020)). The search yielded 187 patents, of which only 18 described anthelminthic treatment methods using EOs or their components, and only 6 patents were granted (Table 3). This search is not exhaustive, as patents solely mentioning the names of EO components would not be included. Nevertheless, this overview provides some insights into the research efforts and attempts at commercialization of EO (components). Moreover, the extent to which natural products can be patented is limited, which would lead to fewer patent applications [203].

The therapeutic use of the EOs of *Kunzea ambigua* against intestinal parasites has been patented in the EU and Australia [204,205]. Although the patent document describes the conduction of clinical trials with the EO, we found no evidence to support its anthelmintic efficacy. Moreover, its efficacy as an insecticide has been disproven [206,207]. On the other hand, geraniol, patented for agricultural use in South Korea [208], exhibits anthelmintic activity against *C. elegans* in vitro [209].

Among the EOs and components listed in patent KR 100960871 B1 [210], *Trachyspermum ammi* EO, thymol and carvacrol have previously demonstrated anthelmintic activity [141,177,211]. Apart from direct anthelmintic activity, EOs play alternative roles in half of the patents granted. For example, cinnamon EO has been patented as a synergist of abamectin B2, a known anthelmintic [212]. Additionally, tea-tree EO and the EO components 1,8-cineole, 1,4-cineole, eugenol, limonene and citronellol are patented as part of a solvent system for benzimidazole anthelmintics [213,214,215,216]. Patent US 8968798 B2 also lists several EOs and EOC as additives to a mixture of isobutyric acid and isobutyric anhydride produced to combat plant pathogens [217,218,219]. Although this review focuses strongly on the direct anthelmintic activity of EOs, their alternative uses as part of solvent systems or as synergists provide interesting avenues for further research, which may lead to patentable findings.

## 10. Future Prospects and Conclusions

Infection, mortality and disability due to nearly 77,000 species of helminthic parasites and some 44,000 host species are extremely high in developing countries, and they form a major share of NTDs. A range of nematodes and trematodes are the major human helminthic parasites. Moreover, all the vertebrate groups are affected by them with significant losses of production, particularly in livestock. The existing anthelmintic assay methods, such as fecal egg count and others, are also not necessarily predictive of therapeutic efficacy.

The current approach to controll helminthic infections is mostly based on preventive pharmacotherapies. However, this has disadvantages, such as drug resistance, adverse effects, undesirable toxicity and recurrence. It is evident plant-derived EOs and some of their components have considerable potential for safe treatment and prevention of helminthic disease. The major EO components, such as monoterpenes (limonene—28–73%, gamma-terpene—10–35%, *p*-cymene, α-pinene—24–27%), sesquiterpenes (β-caryophyllene—10–36%, caryophyllene—50.26%), alcohols (citronellol—37.7%, geraniol—53–81%, terpinen-4-ol, β-linalool—73.2%), EO-phenol (thymol—50–60%, carvacrol—40–94%), phenyl methyl ethers (eugenol—43–53%), aldehydes (citral, citronellal), esters (linalool acetate, geranyl acetate), ketones and EO quinones (thymoquinone—42%), which occur in diverse plant species, have proven effective against a range of helminthic parasites.

However, only limited in vivo studies are available thus far. The cost of clinical development (USD 50–100 million for animal health products and over USD 2.5 billion for human drugs) presents a significant hurdle for compounds with poor prospects for patent protection [220]. Moreover, the low cost of current anthelmintics requires new drugs to be equally inexpensive, unless they are therapeutically superior. In addition, the mechanism of action of such components remains largely unknown, which impedes registration. Resistance is increasing, but in many regions, not yet to the point where treatment with established anthelmintics has become futile. Moreover, newer drugs to which no resistance has yet developed risk being kept for the relatively small fraction of resistant cases, which would further limit their commercial potential considerably. EOs and their components do not fit well into the classical drug discovery paradigm, since they are complex mixtures of natural products. Some EO components could, however, serve as lead compounds to start a more traditional lead optimization effort.

On the other hand, EOs possess some unusual properties, which may prove advantageous for therapy. The lipophilic nature of EOs allows them to cross the membranes of parasites, as well as the blood–brain barrier. In addition, EOs induce oxidative stress in parasites, increase the levels of nitric oxide in the infected host, reduce parasite resistance to reactive oxygen species and increase lipid peroxidation; all these effects ultimately lead to serious damage to cell membranes.

However, this prospect is hindered due to limited in vivo studies. Moreover, the mechanism of action of such components remains largely unknown. More in vitro and in vivo studies on the mechanisms of action are essential to establish the efficacy of EOs. In addition, most of the studies are focused on the major constituents of EOs, but minor constituents and the synergy of action among various components will be an exciting area of future research. Bioassay-guided purification and identification of constituents using LC-MS and development of authentic standards are promising. Application of nanotechnology (nano-emulsion preparation with the EOs) may help solve problems, such as solubility, release and permeation, bio-availability, odor and bio-destruction, which restrict the therapeutic use of EOs in biological systems.

## Figures and Tables

**Figure 1 molecules-27-08327-f001:**
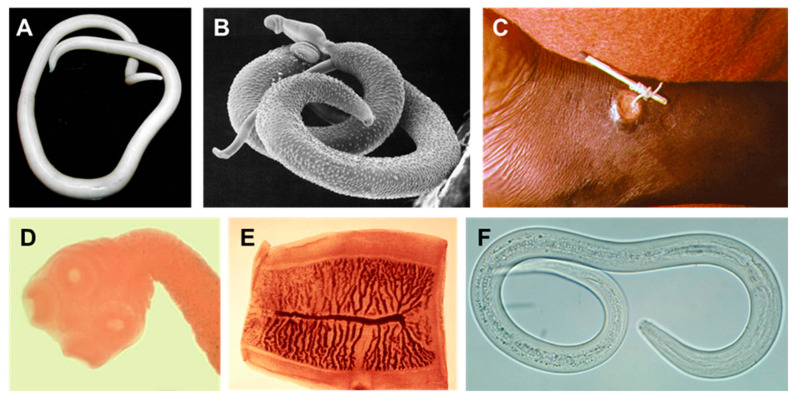
Examples of human helminths. (**A**) Female *Ascaris lumbricoides* (large human roundworm). (**B**) *Schistosoma mansoni* male and female. The smaller female resides in the tegumental fold of the male. (**C**) *Dracunculus medinensis* (Guinea worm) emerging from a human ankle. (**D**,**E**) *Taenia saginata* (beef tapeworm) scolex (head) and proglottid segment. (**F**) *Ancylostoma duodenale* (one of the two species of hookworm). All images reside in the public domain.

**Figure 2 molecules-27-08327-f002:**
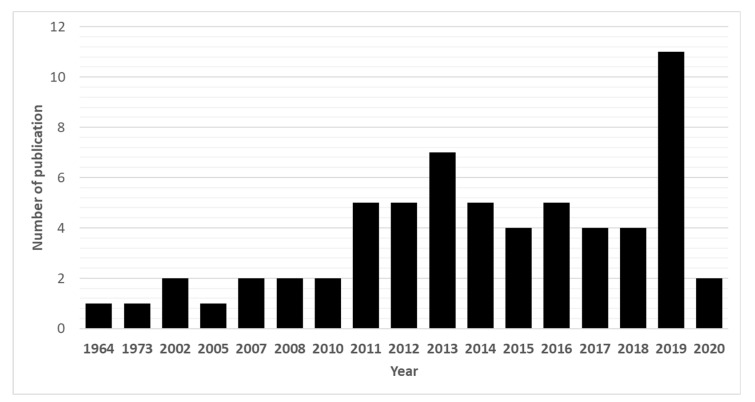
Annual number of anthelmintic papers published from 1964 to 2020 according to PubMed—situation as of July 2020.

**Figure 3 molecules-27-08327-f003:**
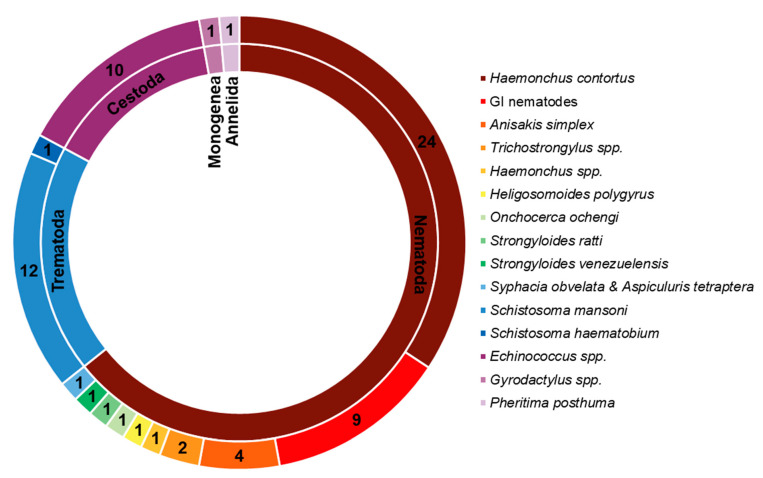
Distribution of species and their taxonomy in the retrieved publications. Cestoda, Trematoda and Monogenea belong to the phylum Platyhelminthes.

**Figure 4 molecules-27-08327-f004:**
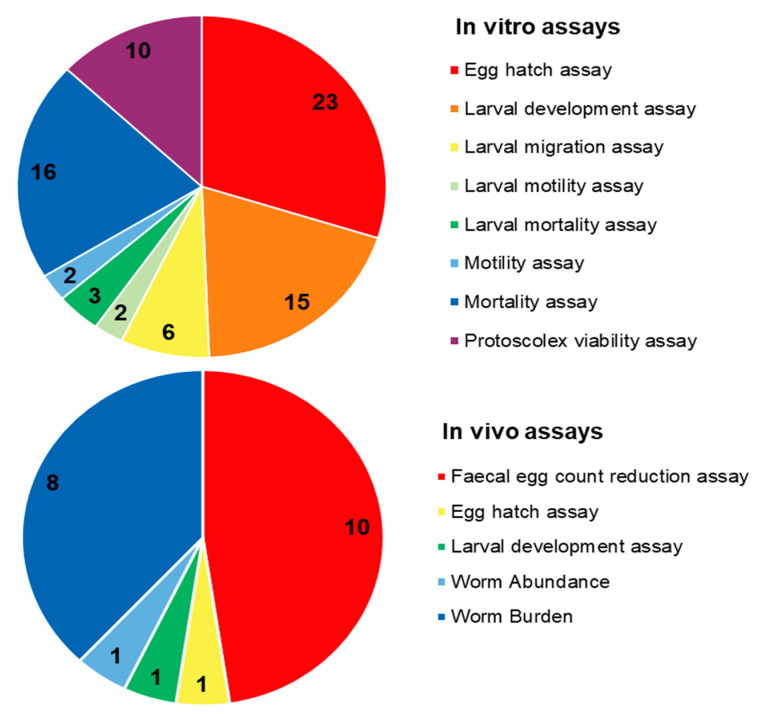
Distribution of different anthelminthic assay methods in in vitro and in vivo experiments.

**Table 1 molecules-27-08327-t001:** List of selected essential oils and their major constituents with in vivo models (for an expanded list, see Appendix A).

Name of the Essential Oil	Major Active Component(s)	Parasite Model	Host	In Vivo	Effect/ Reduction/ Mortality	Assay	References
*Artemisia campestris* L.	β-Pinene (36.4%), 2-Undecanone (14.7%) and Limonene (10.57%)	HP	Mice	5000 mg/kg	72%	FECRA	[129]
*Artemisia campestris*	β-Pinene (36.4%), 2-Undecanone (14.7%) and Limonene (10.57%)	HP	Mice	5000 mg/kg	72%	WB	[129]
** Croton zehntneri* Pax and K.Hoffm.	Anethole (63.88%) and Estragole (21.84%)	SOAT	Albino Swiss mice	800 mg/kg	11.64 ± 35.0%	WB	[130]
* *Cymbopogon citratus* (DC.) Stapf	3,7-Nonadien-2-one, 4,8-dimethyl (24.86%), Geranial (18.98%) and Neral (17.77%)	GIN	Sheep	500 mg/kg	46.90%	WB	[131]
*Cymbopogon citratus*	Geranial (57.3%) and Neral (40.4%)	Hc	Gerbil	800 mg/kg	38.60%	WB	[132]
*Cymbopogon schoenanthus* (L.) Spreng.	Geraniol (59.42%) and Geranial (13.49%)	Hc	Sheep	180 mg/kg	97.50 ± 0.66%	EHA	[133]
*Cymbopogon schoenanthus*	Geraniol (59.42%) and Geranial (13.49%)	Hc	Sheep	180 mg/kg	93.33 ± 1.99%	LDA	[133]
*Corymbia citriodora* (Hook.) K.D.Hill and L.A.S.Johnson (Synonym: *Eucalyptus citriodora* Hook.)	Citronellal (63.94%)	GIN	Sheep	500 mg/kg	41.8%/69.5%	FECRA	[134]
*Eucalyptus citriodora*	Citronellal (67.5%)	GIN	Sheep	500 mg/kg	55.9%/34.5%	FECRA	[135]
*Eucalyptus citriodora*	β-Citronellal (71.77%)	GIN	Goat	500 mg/kg	66.25/60.34/58.45%	FECRA	[136]
*Eucalyptus staigeriana* F.Muell. ex F.M.Bailey	Limonene (72.91%)	GIN	Sheep	365 mg/kg	60.79%	WB	[137]
*Eucalyptus staigeriana*	Limonene (28.82%), E-Citral (14.16%) and Z-Citral (10.77%)	Hc	Goat	500 mg/kg	61.4/76.57/73.66%	FECRA	[138]
* *Lippia sidoides* Cham.	Thymol (59.65%) and (E)-Caryophyllene (10.60%)	SOAT	Albino Swiss mice	1600 mg/kg	68.94 ± 15.1%	WB	[130]
*Lippia sidoides*	Thymol (59.65%) and (E)-Caryophyllene (10.60%)	Ts	Sheep	200 mg/kg	63.6 ± 10.2%	WB	[139]
*Lippia sidoides*	Thymol (59.65%) and (E)-Caryophyllene (10.60%)	Hs	Sheep	283 mg/kg	56.9 ± 10.7%	WB	[139]
*Melaleuca alternifolia* (Maiden and Betche) Cheel	Terpinen-4-ol (41.98%), γ-Terpinene (20.15%) and α-terpinene (9.85%)	Hc	Gerbil	0.75 mL/kg	46.36%	WB	[140]
* *Thymus vulgaris* L.	Thymol (50.22%) and para-Cymene (23.76%)	Hc	Sheep	300 mg/kg	/	FECRA	[141]

*—Essential oil selected based on traditional information; Hp—*Heligosomoides polygyrus polygyrus*; SOAT—*Syphacia obvelata* and *Aspiculuris tetraptera*; GIN—Gastro-intestinal nematodes; Hc—*Haemonchus contortus*; Hs—*Haemonchus* spp.; Ts—*Trichostrongylus* spp.; FECRA—Fecal egg count reduction assay; WB—Worm burden; EHA—Egg hatch assay; LDA—Larval development assay; In vivo (95% confidence interval in brackets; or ± SE); Effect/Reduction/Mortality (95% confidence interval in brackets; or ± SE); /− Data not available.

**Table 2 molecules-27-08327-t002:** List of selected essential oils and their major constituents with in vitro activity that were not studied so far with in vivo models (for an expanded list, see Appendix A).

Name of the Essential Oil	Major Active Constituents	Parasite Model	In Vitro	Effect/Reduction/Mortality	Assay	References
*Ageratum conyzoides* (L.) L.	Precocene I (74.30%) and (E)-Caryophyllene (14.23%)	Sm	100 µg/mL	25%/0%	MA	[142]
*Alpinia zerumbet* (Pers.) B.L.Burtt and R.M.Sm.	1,8-Cineole (24.69%), p-Cymene (22.56%) and 4-Terpineol (17.43%)	Hc	3880 (2940–5090) µg/mL	EC50	LDA	[143]
*Alpinia zerumbet*	1,8-Cineole (24.69%), p-Cymene (22.56%) and 4-Terpineol (17.43%)	Hc	940 (670–1280) µg/mL	EC50	EHA	[143]
*Anethum graveolens* L.	Dihydrocarvone (39.1%), Carvone (22.24%) and D-Limonene (16.84%)	Hc	6 µg/mL	IC50	EHA	[144]
* *Anthemis aaronsohnii* Eig (Synonym: *Anthemis nobilis* L.)	Isobutyl angelate (29.26%) and Isoamyl angelate (15.27%)	Hc	117 µg/mL	IC50	LDA	[145]
*Arisaema franchetianum* Engl.	Linalool (8.89%)	Hc	1100 (940–1270) µg/mL	CE50	LDA	[146]
*Arisaema lobatum*	Carvacrol (7.05%)	Hc	480 (390–570) µg/mL	CE50	LDA	[146]
*Artemisia lancea* Vaniot	1,8-Cineole (34.56%) and Camphor (16.65%)	Hc	1430 (1040–1840) µg/mL	LC50	LMGA	[147]
*Artemisia vulgaris* L.	Caryophyllene (37.45%), Germacrene D (16.17%) and Humulene (13.66%)	Hc	1200 µg/mL	LC50	LMGA	[148]
*Baccharis dracunculifolia* DC.	(E)-Nerolidol (33.51%) and Spathulenol (16.24%)	Sm	10 µg/mL	100%	MA	[149]
*Baccharis trimera* (Less.) DC.	Germacrene D (15.31), trans-Caryophyllene (14.77%) and Bicyclogermacrene (14.67%)	Sm	130 µg/mL	80%	MA	[150]
*Bunium persicum* (Boiss.) B.Fedtsch.	γ-Terpinene (46.1%) and Cuminaldehyde (15.5%)	Es	25 µL/mL	100/100%	PVA	[151]
* *Citrus aurantifolia* (Christm.) Swingle	Limonene (56.37%), β-Pinene (11.86%) and γ-Terpinene (11.42%)	Hc	44 µg/mL	IC50	LDA	[145]
*Citrus limon* (L.) Osbeck	Limonene (29.9%) and β-Pinene (12.0%)	Sm	81.7 µg/mL	LC50	MA	[152]
*Coriandrum sativum* L.	β-Linalool (73.21%)	Hc	2890 (2600–3200) µg/mL	EC50	LDA	[143]
* *Cosmos sulphureus* Cav.	2,6-di-tert-butyl-4-methylphenol (44.98%), Germacrene D (33.70%) and β-Caryophyllene (10.23%)	Sm	100 µg/mL	75%/50%	MA	[153]
*Curcuma longa* L.	β-Turmerone (21.8%), Ar-Turmerone (14.7%) and α-Turmerone (12.4%)	Es	200 µL/mL	100%	PVA	[154]
*Cymbopogon martini* (Roxb.) W.Watson	Geraniol (81.4%) and Isomenthyl isomenthyl acetate (10.1%)	Ts	130 (110–140) µg/mL	CL50	EHA	[155]
*Cymbopogon martinii*	Geraniol (69.63%)	Pp	20,000 µg/mL	3.21 ± 0.31 min	MA	[156]
** Dysphania ambrosioides* (L.) Mosyakin and Clemants	cis-Piperitone oxide (35.2%), para-Cymene (14.5%) and trans-Isoascaridole (14.1%)	Sm	6.50 ± 0.38 µg/mL	LC50	MA	[157]
*Ferula gummosa* Boiss.	β-Pinene (57.0%) and β-Acorenone (11.4%)	Es	17.18 µg/mL	LC50	PVA	[158]
* *Foeniculum vulgare* Mill.	(E)-Anethole (69.8%) and Limonene (22.5%)	Sm	100 µg/mL	50 ± 25%/50 ± 25%	MA	[159]
** Lavandula angustifolia* Mill. (Synonym: *Lavandula officinalis* Chaix)	Linalool acetate (35.97%) and trans-Sabinene hydrate (29.17%)	Hc	280 µg/mL	IC50	LDA	[145]
*Mentha piperita*	Menthol (42.5%) and Menthone (27.4%)	Ts	260 (230–300) µg/mL	CL50	LDA	[155]
* *Mentha x villosa* Huds.	Rotundifolone (70.96%)	Sm	100 µg/mL	100%	MA	[160]
*Myrtus communis* L.	α-Pinene (24.7%), 1,8-Cineole (19.6%) and Linalool (12.6%)	Es	100 µL/mL	100%	PVA	[161]
** Newbouldia laevis* (P.Beauv.) Seem.	β-Caryophyllene (36.08%)	Sr	51.7 ± 7.7 µg/mL	IC50	LMGA	[162]
*Nigella sativa* L.	Thymoquinone (42.4%), para-Cymene (14.1%) and Carvacrol (10.3%)	Es	100 µg/mL	21.6%/76.6%	PVA	[163]
*Ocimum gratissimum* L.	Eugenol (43.7%) and 1,8-Cineole (32.71%)	Hc	0.5vol%	100.0 ± 6.13%	EHA	[164]
*Origanum compactum* Benth.	Carvacrol (50.32%), Thymol (14.8%) and γ-terpinene (13.6%)	As	429 µg/mL	LD50	MA	[165]
*Origanum compactum*	Carvacrol (59.1%) and para-Cymene (11.7%)	Sh	1 µg/mL	100%	MA	[166]
* *Origanum syriacum* L.	Carvacrol (82.6%)	As	87 µg/mL	LC50	LMA	[167]
*Pelargonium radens* H.E.Moore	Citronellol (37.7%), Geraniol (17.6%) and Citronellyl formate (11.0%)	Es	8.52 µg/mL	LC50	PVA	[158]
* *Pinus nigra* subsp. *pallasiana* (Lamb.) Holmboe	α-Pinene (27.46%) and β-Caryophyllene (11.03%)	Es	10,000 µg/mL	61.69%/81.76%	PVA	[168]
*Piper aduncum* L.	Dillapiole (76.2%)	Hc	100 (90–110) µg/mL	IC50	LDA	[169]
*Plectranthus neochilus* Schltr.	β-Caryophyllene (28.23%), α-Pinene (12.63%) and α-Thujene (12.22%)	Sm	100 µg/mL	100%/100%	MA	[170]
*Rosmarinus officinalis* L.	1,8-Cineole (42.11%), 2-Bornanone (16.37%) and α-Pinene (14.76%)	GINS	7100 µg/mL	97.40%	EHA	[171]
*Ruta chalepensis* L.	2-Nonanone (25.31%), 2-Undecanone (24.01%) and Limonene (12.82%)	GINS	1290 ± 1100 µg/mL	EC50	LMGA	[172]
* *Satureja khuzistanica* Jamzad	Carvacrol (94.97%)	Es	10,000 µg/mL	100.00%	PVA	[173]
*Tagetes minuta* L.	Piperitone (86.27%) and Limonene (13.73%)	Hc	1670 (1020–2530) µg/mL	EC50	LDA	[143]
*Tagetes patula* L.	Piperitenone (23.5%) and Piperitone (20.1%)	Hc	40 (35–44) µg/mL	LC50	LDA	[174]
* *Tanacetum vulgare* L.	β-Thujone (84.13%)	Sm	200 µg/mL	100%	MA	[175]
*Tetradenia riparia* (Hochst.) Codd	Fenchone (18.9%), (E,E)-Farnesol (17.7%) and Aromadendrene oxide (17.3%)	Sm	100 µg/mL	100%	MA	[176]
*Trachyspermum ammi* (L.) Sprague	Thymol (50.07%), γ-Terpinene (23.92%) and para-Cymene (22.9%)	Es	10,000 µg/mL	100.00%	PVA	[177]
* *Zanthoxylum bungeanum* Maxim.	Borneol (18.61%) and β-Elemene (10.87%)	Hc	3980 (2890–5310) µg/mL	LC50	EHA	[178]
* *Zanthoxylum zanthoxyloides* (Lam.) Zepern. and Timler	γ-Terpinene (18.0%) and Undecane (14.84%)	Sr	18.2 ± 0.5 µg/mL	IC50	EHA	[162]

*—Essential oil selected based on traditional information; Sm—*Schistosoma mansoni*; Hc—*Haemonchus contortus*; ES—Hydatid cyst (*Echinococcus* spp.) protoscolices; GINC—GI nematodes (cattle); Ts—*Trichostrongylus* spp.; Pp—*Pheritima posthuma*; Sr—*Strongyloides ratti*; As—*Anisakis simplex*; GINS—Gastro-intestinal nematodes (sheep); MA—Mortality assay; LDA—Larval development assay; EHA—Egg hatch assay; LMA—Larval mortality assay; LMgA—Larval migration assay; PVA—Protoscolex viability assay; In vitro (95% confidence interval in brackets; or ± SE); Effect/Reduction/Mortality (95% confidence interval in brackets; or ± SE).

**Table 3 molecules-27-08327-t003:** Patented anthelmintic essential oils or essential oil components as of July 2020.

EO/EO Component	Role	Other Components	Publication Number	Application Year	Title	Institution/ Applicant
Cinnamon EO	Synergist	Abamectin B2	CN 107372644 B	2017	Pesticide for controlling nematodes	Wuhan Kernel Biotech Co., Ltd.
**Geraniol**	Main component	Ganghal extract	KR 101862202 B1	2016	Nematicidal composition containing geraniol	Ecowin Co., Ltd.; Kyung-Bon, Koo
Basil EO, black pepper EO, **(+/−) camphor, carvacrol, trans-cinnamaldehyde**, cinnamon leaf EO, cinnamon bark EO, **citronellol, citral, (+/−) citronella**, clove bud EO, **eucalyptol**, eucalyptus EO, **eugenol**, fennel EO, **geraniol**, ginger EO, jojoba EO, lemongrass EO, **limonene, linalool**, patchouli EO, peppermint EO, **α-terpinene**, rosemary EO, tea-tree EO and thyme EO	Additional component	isobutyric acid, isobutyric anhydride, napthalene and/or caryophyllene	CA 2757537 C; EP 2413692 B1; US 8968798 B2	2010; 2010; 2010; 2013	Compositions of volatile organic compounds and methods of use thereof	Synthetic Genomics Inc.
**1,8-cineole, 1,4-cineole, eugenol, limonene**, tea-tree EO, **citronellol**	Solvent system	Benzimidazole (triclabendazole) and a lactone solvent (γ-hexalactone, moxidectin)	AU 2009245834 B2; CA 2737102 C; CN 102176899 B; EP 2331068 B1; JP 5547738 B2; KR 101318603 B1; RU 2493825 C2; US 9283176 B2; ZA 201103282 B	2009; 2009; 2009; 2009; 2009; 2009; 2009; 2009; 2011	Benzimidazole anthelmintic compositions	Zoetis Services LLC (AU, EU, JP, US) Wyeth LLC (CA, CN, SK, RU) Pah W LLC (ZA)
Ajowan EO, styrax EO or ballerina EO; thymol**, carvacrol, trans-** **cinnamyl alcohol** and **cis-asaron**	Main component		KR 100960871 B1	2008	Composition of nematocides comprising plant essential oils	The Republic of Korea (Forestry Administration Forestry Research Institute)
*Kunzea ambigua* EO	Main component		AU 2008241370 B2; EP 2192912 B1	2008; 2008	Essential oil of *Kunzea ambigua* and methods of use	Hood, John James David

## Data Availability

Not applicable.

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
