# Peer review of "Essential Oils as Novel Anthelmintic Drug Candidates"

_molecules, 2022, doi:10.3390/molecules27238327_

Round 1

Reviewer 1 Report

Few corrections: Remove the already canceled "additinally" Page 7, line297;

please correct the name of 3,7-nonadiene, 4,8-dimethyl on page 23, second paragraph; omit the dot after (10.1%) on page 26 and make all letters italic of Anthemis nobilis on page 26; The sentence at the beginning of the third paragraph on page 29 "On the basis.." should be re-written and is hardly understandable..Last paragraph page 29: "The authors able.." should be changed to "Ther authors were able"

Author Response

Remove the already canceled "additinally" Page 7, line297;

Authors’ response: Done

please correct the name of 3,7-nonadiene, 4,8-dimethyl on page 23,

Authors’ response: Done

second paragraph; omit the dot after (10.1%) on page 26 and make all letters italic of Anthemis nobilis on page 26;

Authors’ response: Done

The sentence at the beginning of the third paragraph on page 29 "On the basis.." should be re-written and is hardly understandable.

Authors’ response:

Last paragraph page 29: "The authors able.." should be changed to "Ther authors were able"

Authors’ response: Done

Reviewer 2 Report

The manuscript presents very interesting and practical information, congratulations to the authors. There are several points needing revision, especially concerning the correct designation of infection by helminths (helminthosis, helminthoses, instead of helminthiasis or helminthiases). In the same way, parasitic infections must be written as fasciolosis or teniosis (ending with -osis).

Other items are:

Introduction

Line 30, 34: The term “helminthiasis” is incorrect, it should be changed by “helmintosis” if caused by one helminth species, or “helminthoses” if two or more.

L39: “schistosomiasis” is incorrect, the correct is “schistosomosis”.

L43: “helminthic biology” is not adequate, please replace by “helminths biology”.

L49: “chemotherapy” is highly associated to describe treatment of cancer, and here it should be better “deworming” or “anthelmintic treatment”.

L158: Replace “Infection” by “infection”.

L172: As mentioned before, replace “helminthiases” by “helminthoses”. It appears also that “helminthiasis” is used as synonym of “helminthiases”…

L202: It is required to provide information on different biological control strategies based on the use of soil saprophytic fungi (Mucor circinelloides, Pochonia chlamydosporia, Duddingtonia flagrans…) to reduce the presence and survival of free-living stages (eggs, larvae), avoiding thus infection.

L358: Essential oils are abbreviated as “EOs”, but in L383 as “Eos”, please uniform it.

Page 28: The sentence “Traditional knowledge the key to find novel anthelmintic drug candidates” appears incomplete.

Pages 29-30: It seems that EO is indistinctly associated to one or more essential oils. Please, revise it.

Tables

Please, try to show the scientific names complete, and not into two lines separated by a hyphen. Table 2 is especially hard to read due to not only scientific names, but also Effect/Reduction/Mortality values are in different lines, which makes very difficult to interpret data shown. And the same applies in Table 3. Authors are strongly encouraged to improve the Tables.

Author Response

The manuscript presents very interesting and practical information, congratulations to the authors. There are several points needing revision, especially concerning the correct designation of infection by helminths (helminthosis, helminthoses, instead of helminthiasis or helminthiases). In the same way, parasitic infections must be written as fasciolosis or teniosis (ending with -osis).

Authors’ response: Thank you for appreciating our review as well as for your detailed comments. All suggestions were taken into account and corrected (indicated in red/blue) with track changes.

Other items are:

Introduction

Line 30, 34: The term “helminthiasis” is incorrect, it should be changed by “helmintosis” if caused by one helminth species, or “helminthoses” if two or more.

Authors’ response: Done

L39: “schistosomiasis” is incorrect, the correct is “schistosomosis”.

Authors’ response: Done

L43: “helminthic biology” is not adequate, please replace by “helminths biology”.

Authors’ response: Done

L49: “chemotherapy” is highly associated to describe treatment of cancer, and here it should be better “deworming” or “anthelmintic treatment”.

Authors’ response: Done

L158: Replace “Infection” by “infection”.

Authors’ response: Done

L172: As mentioned before, replace “helminthiases” by “helminthoses”. It appears also that “helminthiasis” is used as synonym of “helminthiases”…

Authors’ response: Done

L202: It is required to provide information on different biological control strategies based on the use of soil saprophytic fungi (Mucor circinelloidesPochonia chlamydosporiaDuddingtonia flagrans…) to reduce the presence and survival of free-living stages (eggs, larvae), avoiding thus infection.

Authors’ response: Done.

L358: Essential oils are abbreviated as “EOs”, but in L383 as “Eos”, please uniform it.

Authors’ response: Done

Page 28: The sentence “Traditional knowledge the key to find novel anthelmintic drug candidates” appears incomplete.

Authors’ response: Done.

Pages 29-30: It seems that EO is indistinctly associated to one or more essential oils. Please, revise it.

Authors’ response: Done

 Tables

Please, try to show the scientific names complete, and not into two lines separated by a hyphen.

Authors’ response: Thank you for your suggestion. In our initial submission these names are not separated and type setter made this.

Table 2 is especially hard to read due to not only scientific names, but also Effect/Reduction/Mortality values are in different lines, which makes very difficult to interpret data shown. And the same applies in Table 3. Authors are strongly encouraged to improve the Tables.

Authors’ response: Done 

Reviewer 3 Report

This manuscript aims to provide a detailed overview of essential oils and their components as anthelmintic treatment against a wider variety of helminths. Advances in the domain have been achieved and the content of the paper is related to the scope of the journal but the authors should consider the below comments to improve their paper.

1. Please insert a list of abbreviations which would be useful for the readers of the paper.
2. Tables 1 and 2 are to long I think it would be adequate to rethink them as well the discussion based on these tables which is quite short considering the length of the tables. Maybe they could be divided in smaller tables, focusing on more strongly related literature data.

3. The structure of compounds should be deleted. I do not see what is the point of showing them..There are no discussion based on the structure of compounds

4. Conclusion and future perspectives need to be revised. For a review article there it is too hazy, it needs more details. Also, the conclusion section needs to include more numerical data.

Author Response

This manuscript aims to provide a detailed overview of essential oils and their components as anthelmintic treatment against a wider variety of helminths. Advances in the domain have been achieved and the content of the paper is related to the scope of the journal but the authors should consider the below comments to improve their paper.

Authors’ response: We really appreciate your valuable time for reviewing our manuscript. We have made the requested changes and highlighted them in red with track changes.

  1. Please insert a list of abbreviations which would be useful for the readers of the paper.

Authors’ response: Abbreviations included.

  1. Tables 1 and 2 are to long I think it would be adequate to rethink them as well the discussion based on these tables which is quite short considering the length of the tables. Maybe they could be divided in smaller tables, focusing on more strongly related literature data.

Authors’ response: Tables are modified and details of the higher version Tables are provided as supplementary file.

  1. The structure of compounds should be deleted. I do not see what is the point of showing them. There are no discussion based on the structure of compounds

Authors’ response: All structures deleted as suggested.

  1. Conclusion and future perspectives need to be revised. For a review article there it is too hazy, it needs more details. Also, the conclusion section needs to include more numerical data.

Authors’ response: This section was expanded and more detail was added, as well as some numerical data.